# Composition and functional diversity of soil and water microbial communities in the rice-crab symbiosis system

Guo Yang[1], Tian Juncang[1,2,3]*, Wang Zhi[4]

1 Institute of Civil Engineering and Water Conservancy Engineering, Ningxia University, China, 2 Ningxia Water Saving Irrigation and Water Resources Control Engineering Technology Research Center, China, 3 Engineering Research Center of Ministry of Education of Modern Agricultural Water Resources Utilization in Dry Area, China, 4 Department of Earth and Environmental Sciences, California State University, Fresno, CA, United States of America

* slxtjc@163.com

**Data Availability Statement:** All relevant data are within the manuscript and its Supporting Information files.

**Funding:** National Key Research and Development Project (No. 2021YFD1900605-04) First-class

## Abstract

Rice-crab co-culture is an environmentally friendly agricultural and aquaculture technology with high economic and ecological value. In order to clarify the structure and function of soil and water microbial communities in the rice-crab symbiosis system, the standard rice-crab field with a ring groove was used as the research object. High-throughput sequencing was performed with rice field water samples to analyze the species and abundance differences of soil bacteria and fungi. The results showed that the OTU richness and community diversity in soil were significantly higher than those in water, while there were significant differences in soil microbial diversity and OTU richness in water sediments. The dominant species at the bacterial phylum level were Amoebacteria, Cyanobacteria, Actinomycetes, Synechococcus and Greenbacteria, and at the genus level the dominant species were *norank_f_norank_o_Chloroplast*, *unclassified_f_Rhodobacteraceae*, *LD29*, *Cyanobium_PCC-6307*, and *norank_f_MWH-UniP1_aquatic_group*. The dominant species at the fungal phylum level are unclassified_k_Fungi, Ascomycota, Rozellomycota, Phaeomycota and Stenotrophomonas, and at the genus level the dominant species are *unclassified_k_Fungi*, *unclassified_p_Rozellomycota*, *Metschnikowia*, *Cladosporium*, *unclassified_p_Chytridiomycota*. The dominant phylum may rely on mechanisms such as organic matter catabolism, secretion of secondary metabolites and phototrophic autotrophy, as predicted by functional gene analysis. The main functional genes are related to metabolic functions, including secondary product metabolism, energy metabolism, and amino acid metabolism.

## Introduction

Rice is a vital crop in China, serving not only as a major food source but also fulfilling important wetland ecosystem functions such as pollution absorption and control. It plays a crucial role in ensuring food security and maintaining the stability of agricultural ecosystems [1]. The integrated farming system of rice and eco-friendly aquatic animals, such as crabs, fish, shrimp,

discipline of Ningxia High Education Institutions (Water Engineering Discipline) funded project (Grant No. NXYLXK2021A03). The funder TIAN Juncang provided financial support for the experiment. The specific contributions of the author are as follows: Data curation: Yang Guo. Formal analysis: Yang Guo. Methodology: Juncang Tian, Zhi Wang Resources: Juncang Tian. Supervision: Juncang Tian. Writing – original draft: Yang Guo. Writing – review & editing: Juncang Tian.

**Competing interests:** The authors have declared that no competing interests exist.

frogs, and ducks, is increasingly becoming a focus of attention, particularly the rice-crab farming system. From an ecological perspective, the rice-crab model forms a mutually beneficial agricultural ecosystem. The rice provides a cool environment and abundant food for the crabs, while the crabs' activities help aerate the soil and control weeds. Moreover, their waste acts as a natural fertilizer for the rice, significantly reducing the need for chemical fertilizers and pesticides. This natural cycle not only enhances soil nutrient levels, enzyme activity, and microbial biomass carbon and nitrogen but also increases nitrogen use efficiency and reduces nutrient loss [2, 3]. It greatly minimizes the negative environmental impacts of agricultural production [4], including the use of chemical pesticides and fertilizers. Furthermore, crabs, as high-value agricultural products, increase land use efficiency and output, making the rice-crab symbiosis a unique "low-carbon, high-efficiency circular farming model" in the Yinbei Irrigation District.

Soil and water are the foundational components of the rice-crab ecosystem, playing a pivotal role in this complex system. Microbial communities, closely linked to both soil and water ecosystems, play an irreplaceable role in this regard [5]. Soil microbes decompose organic matter in the soil, participate in mineralization processes, and influence soil nutrient use and biogeochemical cycling, ultimately affecting rice growth, quality, and yield. In the rice agroecosystem, water layer microbial communities reflect the conditions of the water source and surrounding soil [6, 7]. Additionally, these microbial communities help regulate the water environment and contribute to water purification, forming a harmonious ecosystem together with rice and crabs.

Current research on the rice-crab symbiosis primarily focuses on rice yield, soil physicochemical properties, crop production efficiency, and greenhouse gas emissions [8, 9]. However, studies on microbial distribution and cycling within the overall ecosystem are relatively limited. It is worth noting that maintaining microbial species diversity is crucial for building a sustainable agricultural production system [10]. Soil and water microbes, especially fungi, play a vital role in nutrient cycling, pollutant degradation, and crop disease prevention [11]. Therefore, strengthening research on microbial communities in the rice-crab symbiosis will help us better understand the mechanisms driving this ecosystem. This study, based on the Illumina sequencing platform, analyzed microbial sequencing of rice rhizosphere soil, trench bottom soil, trench water, and rice field water in a rice-crab co-cultivation area. The study aimed to explore the abundance, composition, and diversity of bacterial and fungal communities at different sites in the rice-crab co-culture system, providing a solid foundation for promoting healthy agricultural aquaculture.

## Materials and methods

### Experimental design

The experimental site is located in Yuxiang Village, Helan County, Yinchuan City, Ningxia Hui Autonomous Region, China (38°40′30″ N, 106°17′56″ E). The soil type at the site is irrigated alluvial soil, with irrigation water sourced from the Yellow River. The experimental site is at an elevation of 1102 m, and the dry bulk density of the topsoil (40 cm) is 1.43 g·cm$^{-3}$, with a field water holding capacity of 22.5%. The basic soil properties include a pH of 8.46, organic matter content of 9.45 mg·kg$^{-1}$, available nitrogen of 32.2 mg·kg$^{-1}$, available phosphorus of 11.6 mg·kg$^{-1}$, and available potassium of 128.0 mg·kg$^{-1}$.

The rice variety used was 'Qiu You 88', which was transplanted on May 25, 2018, and harvested on October 8. Crab larvae were introduced on May 28 and harvested on September 5. The plot was 170 m long and 40 m wide, with row and plant spacing of 10 cm × 5 cm. A 30 cm-high anti-escape fence was set up around the plot. The rice was harvested at maturity on

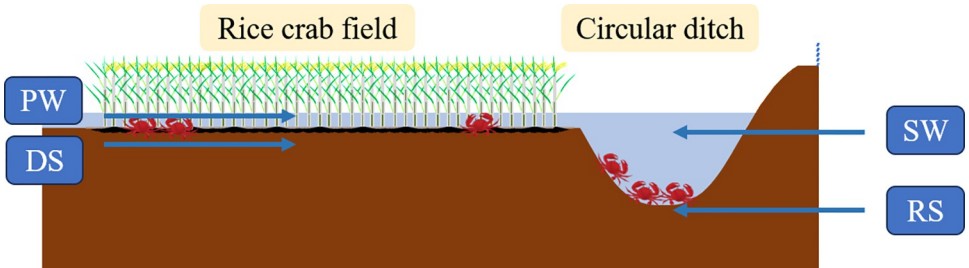

**Fig 1. Sampling schematic diagram.**

October 1. Samples were collected on September 4, 2021, from the rice rhizosphere soil (RS), ditch bottom soil (DS), ditch water (SW), and paddy field water (PW). Each sample was collected from three different locations, mixed thoroughly, and set up with three replicates. The soil samples were cooled and immediately transported to the laboratory, while the water samples were rinsed with sterile water, shaken at 200 rpm for 30 minutes, and filtered through a 0.22 μm membrane. The schematic of the sampling points is shown in Fig 1, and all samples were stored at -80˚C.

## DNA extraction and high-throughput sequencing

Microbial community DNA was extracted using the E.Z.N.A.® Soil DNA Kit (Omega Biotek, Norcross, GA, U.S.) following the manufacturer's instructions. The quality of the extracted DNA was assessed by 1% agarose gel electrophoresis, and the concentration and purity of the DNA were measured using a NanoDrop 2000 spectrophotometer. The bacterial 16S rDNA (V3+V4) regions were amplified using the primer pair 5'-ACTCCTACGGGAGGCAGCA-3' and 5'-GGACTACHVGGGTWTCTAAT-3'. For fungi, the universal ITS2 gene primers 5'-TAGAGGAAGTAAAAGTCGTAA-3' and 5'-TTCAAAGATTCGATGATTCAC-3' were used for amplification. The PCR protocol included an initial denaturation at 95˚C for 3 minutes, followed by 27 cycles of 30 seconds at 95˚C (denaturation), 30 seconds at 55˚C (annealing), and 30 seconds at 72˚C (extension), with a final extension at 72˚C for 10 minutes. The reaction was held at 4˚C after completion (PCR instrument: ABI GeneAmp® 9700).

The PCR reaction system consisted of 4 μL of 5×TransStart FastPfu buffer, 2 μL of 2.5 mM dNTPs, 0.8 μL of each forward and reverse primer (5 μM), 0.4 μL of TransStart FastPfu DNA polymerase, and 10 ng of template DNA, with the final volume adjusted to 20 μL. Each sample was amplified in triplicate. The amplified fragments were purified and recovered using 1% agarose gel electrophoresis, and library construction and sequencing were conducted by Majorbio Biomedical Technology Co., Ltd. (Shanghai, China).

## Bioinformatics analysis

Raw sequencing reads were demultiplexed and filtered for adaptors and quality. Initial denoising was performed to remove sequences with sequencing errors. Paired-end reads were merged based on overlap for each sample using FLASH (v1.2.7) [12], resulting in raw tags. The raw tags were further filtered for quality using Trimmomatic (v0.33), yielding high-quality tag data. Chimera sequences were identified and removed using UCHIME (v4.2), resulting in the final valid data set. OTUs were clustered at 97% similarity using UCLUST in QIIME (v1.8.0) and taxonomically annotated based on the Silva (bacterial) database. Data were normalized according to the sample with the fewest sequences across all samples.

All analyses were conducted on the Majorbio Cloud Platform (https://cloud.majorbio.com). Alpha diversity indices (Ace, Chao, Shannon, Simpson) were calculated using Mothur [13]. PCoA (Principal Coordinate Analysis) based on the Bray-Curtis distance algorithm was used to assess similarities in microbial community structure between samples, and PERMANOVA was applied to test for significant differences in community structure between groups. LEfSe analysis (Linear Discriminant Analysis Effect Size) was used to identify bacterial taxa with significantly different abundances between groups at the phylum to genus levels (LDA > 4, P < 0.05) [14]. Functional prediction of microbial communities was performed using FAPROTAX and FUNGuild [15].

Statistical analysis was performed using the SPSS statistical software package (Version 18.0, SPSS Inc., Chicago, IL, USA), with ANOVA and Duncan's multiple range tests. Differences were considered statistically significant at p < 0.05.

## Results

### Analysis of sequencing results

High-throughput sequencing results (Table 1) showed that a total of 639,489 paired-end reads were obtained from 12 bacterial samples, which generated 574,663 clean tags after filtering, with an average of 47,888 clean tags per sample and an average sequence length of 415 bp. For the 12 fungal samples, a total of 1,138,709 paired-end reads were obtained, and after filtering, 989,311 clean tags were generated, averaging 82,442 clean tags per sample, with an average sequence length of 218 bp. As shown in Table 1, bacterial OTUs were significantly higher than fungal OTUs across the four sample groups, with OTU counts ranging from 272 to 3,498. The Chao index ranged from 282 to 4,102, and the Shannon index ranged from 3.16 to 6.86. All sequencing samples exhibited high coverage rates ($\geq$0.97), indicating that the sequencing results could accurately represent the microbial composition within the samples [16]. Based on richness indices (Chao1 and Ace), bacterial richness from highest to lowest was DS > RS > PW > SW, while the diversity indices (Shannon and Invsimpson) followed the same trend. Fungal diversity was lower than bacterial diversity, with richness decreasing in the order of DS > RS > PW > SW.

Principal Coordinate Analysis (PCoA) based on Bray-Curtis dissimilarity was used to assess overall differences in microbial community structure. The PCoA plot (Fig 2) showed that both bacterial and fungal communities clustered according to their sample sources. For bacterial communities, the first two principal coordinates, PC1 and PC2, explained 88.7% and 7.17% of the total variation, respectively. In contrast, for fungal communities, the first and second axes explained a similar amount of variation, accounting for a total of 81.48%, with p-values of 0.001 for both communities.

**Table 1. Number of OTUs and diversity indices.**

|          | Sample | OTUs      | Shannon    | Simpson        | Ace            | Chao1           | Coverage      |
|----------|--------|-----------|------------|----------------|----------------|-----------------|---------------|
| Bacteria | DS     | 3498±158a | 6.86±0.07a | 0.003±0.0002c  | 4133.65±108.24a | 4101.86±86.32a  | 0.97±0.0004   |
|          | RS     | 3221±249a | 6.74±0.14a | 0.004±0.0012c  | 3763.25±382.15a | 3767.00±339.62a | 0.973±0.003   |
|          | PW     | 1563±89b  | 4.51±0.03b | 0.073±0.0082b  | 2655.83±180.8b  | 2058.57±111.40b | 0.98±0.0010   |
|          | SW     | 1068±89c  | 4.14±0.16c | 0.101±0.0191a  | 2140.02±24.39c  | 1729.46±56.26b  | 0.99±0.0004   |
| Fungi    | DS     | 711±45a   | 3.70±0.18a | 0.07±0.02a     | 742.55±71.19a   | 730.31±73.39a   | 0.997±0.0004  |
|          | RS     | 515±121b  | 3.56±0.66a | 0.12±0.10a     | 525.71±115.37b  | 525.08±106.17b  | 0.999±0.0003  |
|          | PW     | 412±71bc  | 3.35±0.21a | 0.08±0.02a     | 413.53±65.64bc  | 417.44±66.07bc  | 0.999±0.0001  |
|          | SW     | 272±47d   | 3.16±0.08a | 0.09±0.01a     | 278.26±49.42d   | 282.48±53.19d   | 1.000±0.0003  |

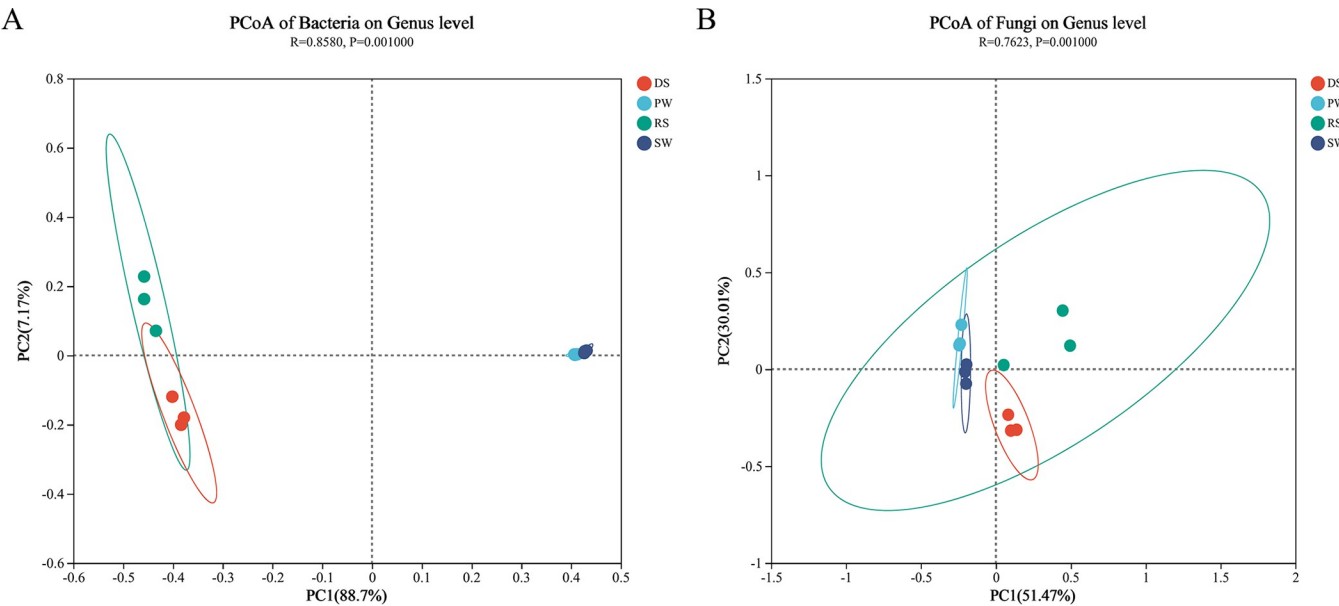

**Fig 2. Principal coordinate analysis (PCoA) analysis of the dissimilarity (Bray-Curtis distance) in microbial communities.** (A) Bacteria. (B) Fungi.

## Composition of microbial communities

In total, 54 phyla were identified across the four sample groups (Fig 3), with 14 phyla exhibiting an abundance greater than 1%. Phyla with an abundance over 10% are considered dominant, while genera with an abundance over 5% are classified as dominant genera. The bacterial communities of DS and RS were similar, with the dominant phyla being Proteobacteria, Bacteroidetes, Actinobacteria, Chloroflexi, and Desulfobacteria. Compared to RS, the relative abundances of Actinobacteria, Chloroflexi, Acidobacteria, and Gemmatimonadetes in DS decreased by 6.15%, 7.85%, 2.01%, and 1.74%, respectively, while the proportions of the other phyla increased. Notably, the relative abundances of Bacteroidetes and Desulfobacteria increased significantly by 7.4% and 4.9%. The PW and SW bacterial communities were also similar, dominated by Proteobacteria, Actinobacteria, Bacteroidetes, and Verrucomicrobia. In PW, the relative abundances of Cyanobacteria, Actinobacteria, Verrucomicrobia, and Vibrio decreased by 5.91%, 0.47%, 1.59%, and 0.21%, respectively, while the proportions of other phyla increased, particularly Proteobacteria and Bacteroidetes, which saw increases of 4.88% and 1.77%.

At the genus level, a total of 1,227 different genera were identified across the four sample groups, with 49 genera exhibiting an abundance greater than 1%. The bacterial communities of DS and RS were similar, but neither had dominant genera exceeding 5%. In DS, the main genera included *norank_f_Bacteroidetes_vadinHA17* (4.06%), *norank_f_steroidobacteraceae* (2.49%), *norank_f_Prolixibacteraceae* (2.36%), and *norank_f_norank_o_SBR1031* (2.11%). In RS, the primary genera were *norank_f_norank_o_SBR1031* (3.97%), *norank_f_norank_o_norank_c_KD4-96* (2.97%), *Marmoricola* (2.95%), *norank_f_Anaerolineaceae* (2.59%), *Nocardioides* (2.39%), *Thiobacillus* (2.20%), *norank_f_norank_o_Vicinamibacterales* (2.14%), and *norank_f_norank_o_Ardenticatenales* (2.04%). In contrast to the soil, the main dominant genera in the PW and SW bacterial communities were *norank_f_norank_o_chloroplast* (27.75%–33.05%) and *Rhodobacteraceae* (5.49%–6.89%). Other relatively abundant genera, such as *LD29*, *Cyanobium_PCC-6307*, *norank_f_MWH-UniP1_aquatic_group*, *Hydrogenophaga*, and *hgcl_clade*, also exceeded 2% in relative abundance.

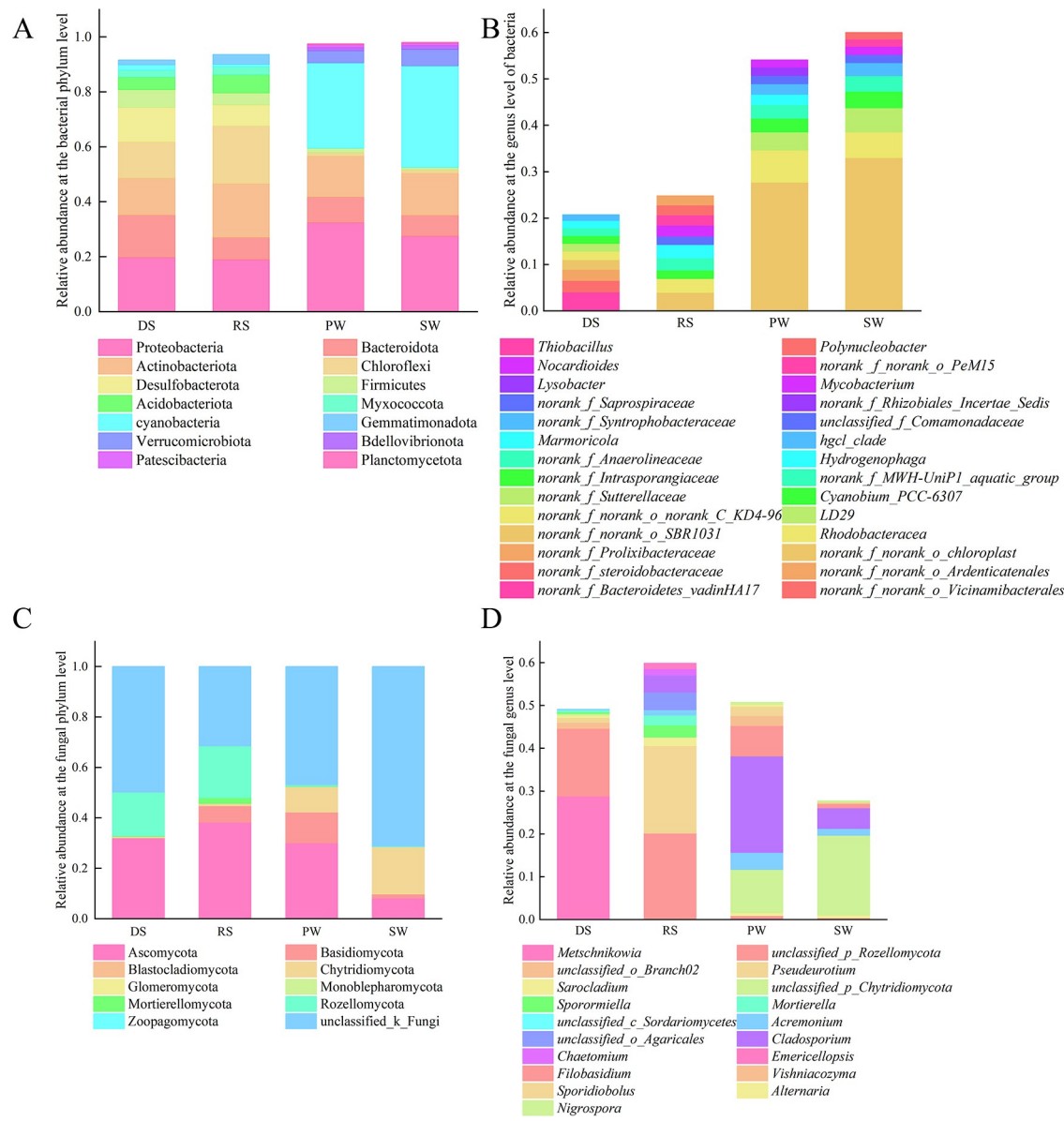

**Fig 3. Relative abundance of microorganisms in the top 10 phyla and genus levels.** (A) Bacterial phylum level. (B) Bacterial genus level. (C) Fungi phylum level. (D) Fungal genus level.

Compared to bacteria, the diversity of fungal communities has decreased. A total of only nine fungal phyla were detected across all samples (Fig 3), alongside a significant number of unclassified fungal phyla. The fungal community composition in the DS and RS was similar, with Ascomycota and Rozellomycota being the dominant phyla. Relative abundances of Basidiomycota, Ascomycota, Rozellomycota, Monoblepharomycota, and Chytridiomycota decreased by 6.41%, 6.17%, 3.23%, 1.97%, and 0.45%, respectively, in DS compared to RS, while the proportions of other phyla changed by less than 0.02%. The fungal community composition in the spring PW and SW was similar, dominated by Ascomycota, Basidiomycota, and Chytridiomycota. In PW compared to SW, the relative abundances of Ascomycota and Basidiomycota increased by 21.98% and 10.57%, respectively, while the relative abundance of Chytridiomycota decreased by 8.57%.

At the genus taxonomic level, a total of 97 genera were identified across the four sample groups, with unclassified genera accounting for 31.46% to 71.32% of the community. As shown in Fig 3, the distribution of fungal genera varied among the four sample groups. The *unclassified_p_Rozellomycota* (15.86% to 20.18%) was a common dominant genus in both DS and RS, along with the dominant genera *Metschnikowia* (28.81%) and *Pseudeurotium* (20.45%) found in these seasons. The unclassified *Chytridiomycota* (10.09% to 18.66%) was the common dominant genus in both PW and SW, while PW also identified the dominant genus *Filobasidium* (7.12%).

## LEfSe analysis of microbial communities

LEfSe analysis was conducted to identify distinct biomarkers with an LDA score greater than 4. These biomarkers exhibited significant changes in the relative abundance of core microorganisms and displayed considerable variability in response to environmental disturbances. As shown in Fig 4, 36 bacterial lineages demonstrated significant differences across all samples (S1 File). The samples from DS, RS, PW, and SW were enriched with different bacterial lineages, showing only 3, 8, 2, and 5 enriched lineages, respectively. At the genus level, DS soil was predominantly enriched with *Steroidobacteraceae*, *Prolixibacteraceae*, and *Bacteroidetes_vadinHA17*, while RS was mainly enriched with *Thiobacillus*, *Vicinamibacterales*, *Marmoricola*, *Nocardioides*, *Anaerolineaceae*, *SBR1031*, *Ardenticatenales*, and *KD4-96*. The PW was characterized by *Rhodobacteraceae* and *Hydrogenophaga*, whereas the SW included *MWH-UniP1_aquatic_group*, *Chloroplast*, *PCC-6307*, *hgcl_clade*, *LD20*, with the most significant taxa being the *Chloroplast* from the phylum Cyanobacteria (LDA score > 5.0). In terms of fungi, DS, RS, PW, and SW each contained only 1, 5, 5, and 2 enriched fungal lineages, respectively. At the genus level, DS soil was mainly enriched with *Metschnikowia*, while RS was characterized by *Pseudeurotium*, *unclassified_p__Rozellomycota*, *Cistella*, *Sporormiella*, and *Mortierella*. PW was dominated by *Cladosporium*, *Filobasidium*, *Acremonium*, *Vishniacozyma*, and *Sporidiobolus*, along with *unclassified_p__Chytridiomycota* and *Rhodotorula* in SW. The results of the LEfSe analysis were generally consistent with the aforementioned microbial community analysis, indicating significant differences in microbial composition among different sampling sites, which reflect the overall cycling effects on microbial communities to some extent.

## Functional prediction of microbial communities

The FAPROTAX function prediction results are shown in Fig 5. The primary taxonomic groups identified were involved in processes such as chlorophyll production, chemoheterotrophy, aerobic chemoheterotrophy, aerobic nitrite oxidation, nitrification, ureolysis, aromatic compound degradation, and hydrocarbon degradation. Notably, one-third of the predicted metabolic functions had a relative abundance greater than 1%. Bacterial OTUs in DS and RS were significantly enriched in aerobic chemoheterotrophy, fermentation, aromatic compound degradation, nitrate reduction, sulfur compound respiration, dark sulfur compound oxidation, sulfate respiration, nitrate respiration, and nitrogen respiration compared to those in PW and SW (p < 0.05). Conversely, PW and SW exhibited a higher abundance of chloroplast-associated, phototrophic, photoautotrophic, cyanobacterial, and aerobic photoautotrophic functional bacterial groups. Additionally, a small proportion of functions related to lignin degradation, aromatic hydrocarbon degradation, aliphatic non-methane hydrocarbon degradation, xylanolysis, and cellulolysis (abundance < 2%) were identified across all samples.

Based on FUNGuild predictions, a total of 296 OTUs were annotated with trophic mode information after excluding unclassified OTUs (Fig 5). The dominant trophic mode across the fungal communities of the four sample groups was saprotrophy. Undefined saprotrophs and

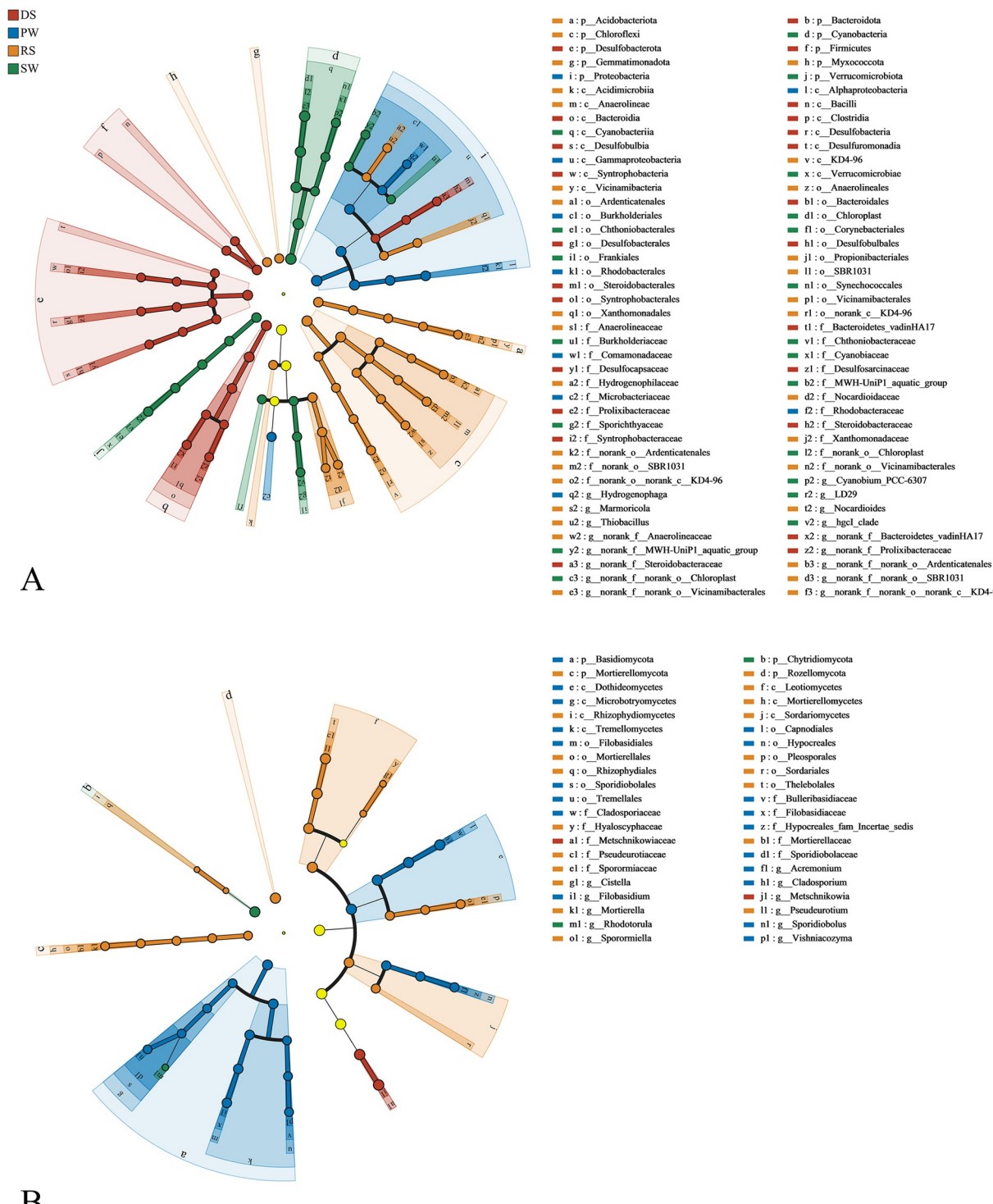

**Fig 4. Linear discriminant analysis (LDA) effect size (LefSe) analysis of soil microbial abundance.** (A) Bacteria. (B) Fungi.

Animal Pathogen-Endophyte-Lichen Parasite-Plant Pathogen-Wood Saprotroph represented 32.78%, 31.16%, 31.81%, and 7.11% of the OTUs in the four sample groups, respectively. The symbiotic mode was primarily composed of arbuscular mycorrhiza, endophytes, and

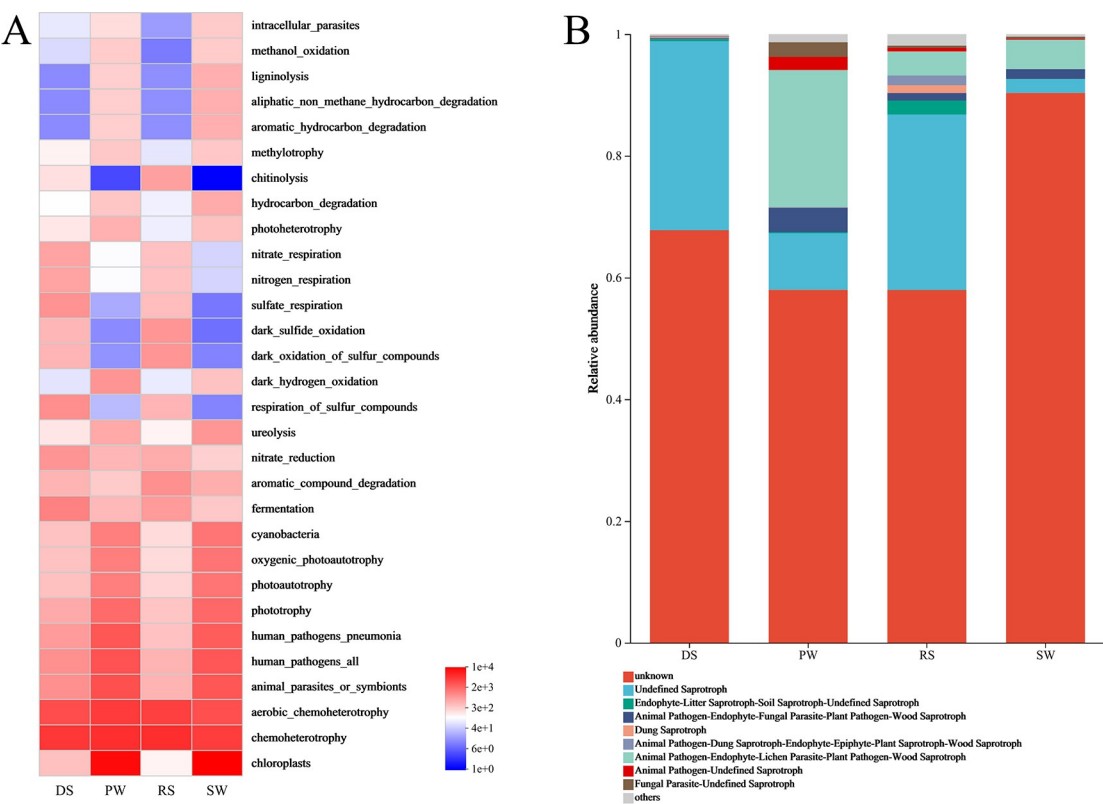

**Fig 5. Functional prediction of microbial communities.** (A) FAPROTAX. (B) FUNGuild.

ectomycorrhiza. RS exhibited the highest number of functional groups, with the Endophyte-Litter Saprotroph-Soil Saprotroph-Undefined Saprotroph (2.30%), Animal Pathogen-Dung Saprotroph-Endophyte-Epiphyte-Plant Saprotroph-Wood Saprotroph (1.61%), and Dung Saprotroph (1.26%) showing higher proportions than in the other three sample groups. In contrast, Animal Pathogen-Endophyte-Fungal Parasite-Plant Pathogen-Wood Saprotroph (4.01%), Animal Pathogen-Undefined Saprotroph (2.16%), and Fungal Parasite-Undefined Saprotroph (2.43%) were more prevalent in PW compared to the other groups.

Based on FAPROTAX annotation, the number of OTUs related to nitrogen cycling is shown in Fig 6. This includes nitrogen fixation, nitrate respiration, nitrogen respiration, nitrous oxide denitrification, denitrification, nitrate respiration, nitrate denitrification, and nitrite denitrification, with the OTU abundance ranked as DS > RS > PW > SW. The relative abundance of methanogenic species is ranked as RS (22.60%) > DS (19.79%) > SW (1.62%) > PW (1.61%).

## Co-occurrence network analysis

To reveal the potential interactions among microorganisms, correlation analyses were conducted on four groups of samples based on the occurrence patterns of operational taxonomic units (OTUs) in the samples (Fig 7). In the bacterial communities, a total of 50 nodes exhibited significant correlations, while the fungal samples showed 48 pairs of strongly correlated nodes. The clustering coefficient reflects the connectivity among neighboring nodes: a clustering coefficient of 1 indicates that all neighboring nodes are fully connected, whereas a coefficient close to 0 suggests minimal connectivity among neighbors. A higher clustering coefficient indicates greater importance of the node.

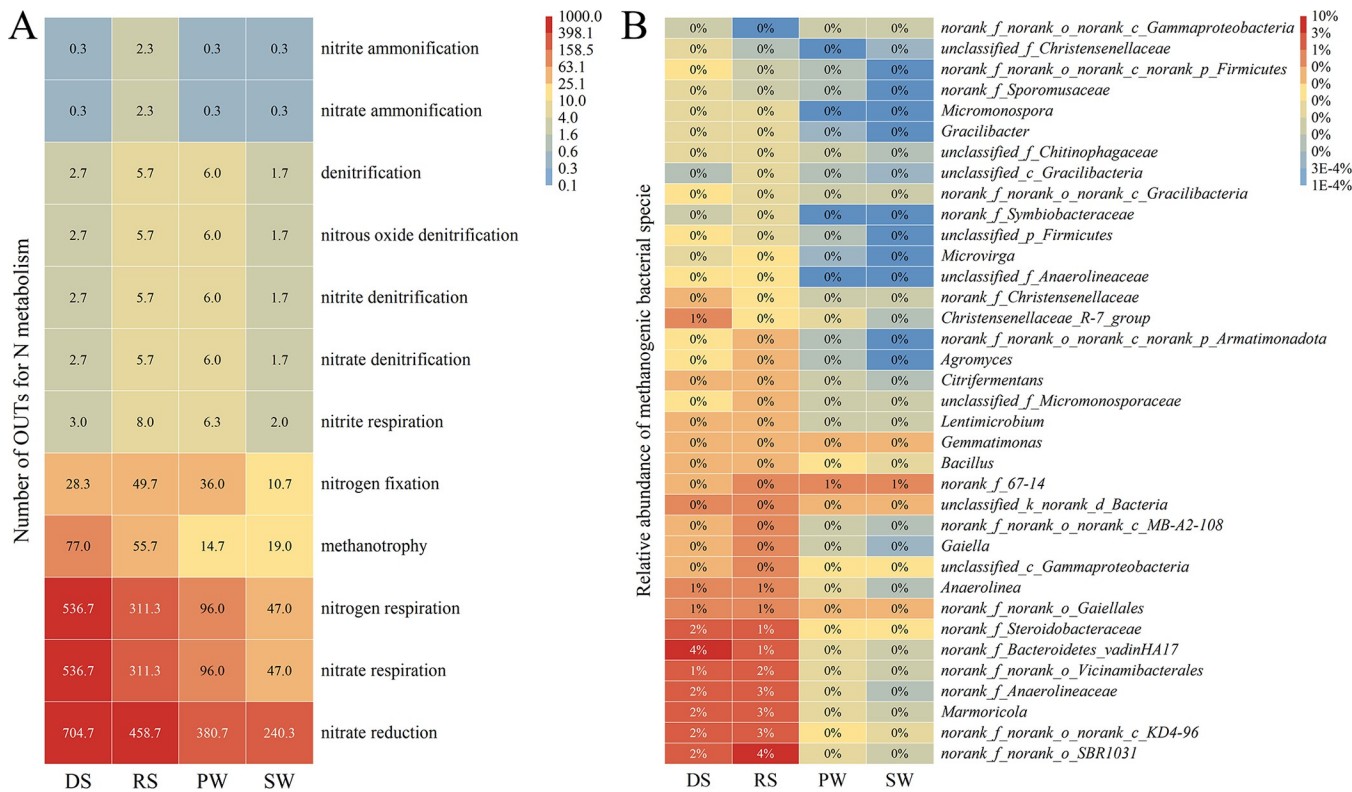

**Fig 6. Number of functional OTUs.** (A) Bacteria involved in N metabolism. (B) Methanogenic bacteria.

In the bacterial correlation network, *norank_f_Rhizobiales_Incertae_Sedis*, *unclassified_f_-Comamonadaceae*, *norank_f_norank_o_Ardenticatenales*, *Algoriphagus*, and *Novosphingobium* were identified as key species. In the fungal correlation network, *Gibberella*, *Hannaella*, *Papiliotrema*, and *Gibellulopsis* played significant roles (Table 2).

## Discussion

### Microbial phylum level community structure and functional composition

In this study, the most abundant bacterial phyla across the majority of the samples were Proteobacteria, Actinobacteria, Bacteroidota, and Chloroflexi, which aligns with the findings of

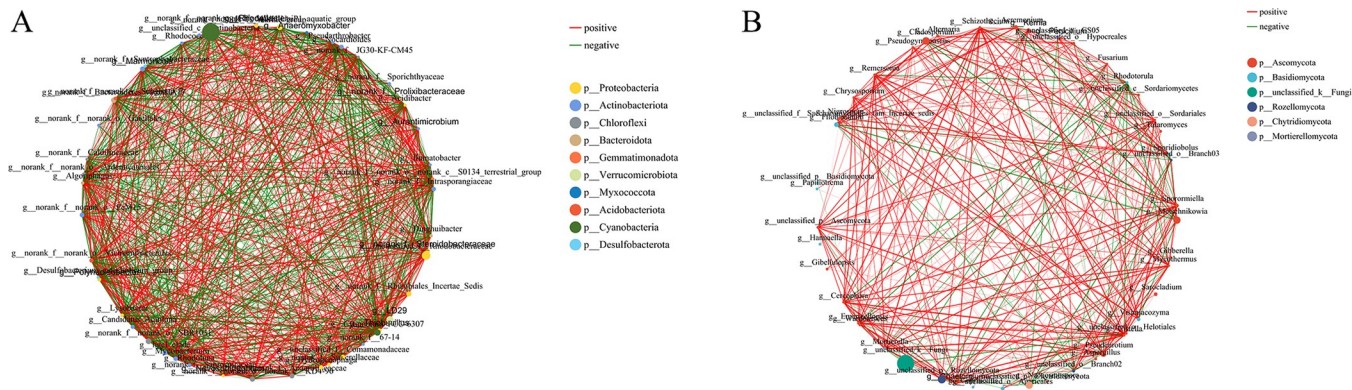

**Fig 7. Network diagram of the correlation between species.** (A) Bacteria. (B) Fungi.

**Table 2. Attributes of the top 15 network nodes.** (left) bacteria. (right) fungi.

| Node | Degree | Clustering coefficient | Node | Degree | Clustering coefficient |
|---|---|---|---|---|---|
| g_norank_f_Rhizobiales_Incertae_Sedis | 38 | 0.933 | g_Gibberella | 3 | 1.000 |
| g_unclassified_f_Comamonadaceae | 40 | 0.927 | g_Hannaella | 4 | 1.000 |
| g_norank_f_norank_o_Ardenticatenales | 41 | 0.924 | g_Papiliotrema | 2 | 1.000 |
| g_Algoriphagus | 39 | 0.920 | g_Gibellulopsis | 5 | 0.900 |
| g_Novosphingobium | 40 | 0.914 | g_Aspergillus | 8 | 0.893 |
| g_unclassified_f_Rhodobacteraceae | 40 | 0.914 | g_Chaetomium | 21 | 0.857 |
| g_Acidibacter | 42 | 0.912 | g_unclassified_o_GS05 | 11 | 0.855 |
| g_Candidatus_Aquiluna | 41 | 0.907 | g_Chrysosporium | 19 | 0.842 |
| g_norank_f_norank_o_PeM15 | 41 | 0.907 | g_unclassified_k__Fungi | 18 | 0.837 |
| g_Mycobacterium | 41 | 0.907 | g_Kernia | 19 | 0.830 |
| g_norank_f_norank_o_Vicinamibacterales | 42 | 0.906 | g_unclassified_o_Branch03 | 14 | 0.791 |
| g_Aurantimicrobium | 38 | 0.895 | g_unclassified_o_Helotiales | 20 | 0.789 |
| g_norank_f_Caldilineaceae | 41 | 0.890 | g_Fusarium | 16 | 0.783 |
| g_Hydrogenophaga | 40 | 0.890 | g_Pseudeurotium | 23 | 0.767 |
| g_Rhodobacter | 41 | 0.884 | g_Emericellopsis | 23 | 0.767 |

Liu et al. [17]. Most Proteobacteria are copiotrophic, characterized by their relatively fast growth rates and ability to utilize a wide range of substrates, and are commonly found in paddy fields [18]. Actinobacteria play a crucial role in the environment by producing hydrolytic enzymes, which help decompose organic matter and collaborate in nutrient cycling to maintain soil biological balance, particularly contributing to the recycling of organic carbon [19]. Chloroflexi, the most dominant phylum in RS, can be classified into phototrophic and non-phototrophic groups, utilizing unstable carbon sources in the environment, breaking down starch, sugars, and peptides, and providing organic acids for other organisms in the sediment, thus playing an indispensable role in the soil carbon cycle [20]. Desulfobacterota, Acidobacteriota, and Firmicutes were dominant in the soil samples, whereas their abundance significantly decreased in water samples, where Cyanobacteria became the dominant phylum.

In terms of fungal diversity, Ascomycota was the most abundant phylum in all samples. The abundance of Rozellomycota in soil samples was notably higher than in water samples. Generally, Rozellomycota are found in freshwater, soil, or marine ecosystems and are often associated with plant pathogenic fungi, with a preference for consuming soil organic matter [21]. Furthermore, they may play an important role in the phosphorus cycle [22]. In water samples, the phylum-level diversity in the PW group was significantly higher than in the SW group. Chytridiomycota had a relative abundance exceeding 10% in all water samples. Chytridiomycota are common aquatic fungi, many of which are saprotrophic, and are active in oxygenated, water-saturated paddy soils. Chytridiomycota are also known parasites, infecting a wide range of organisms, including phytoplankton, zooplankton, fungi, invertebrates, and plants [23]. This indicates that both soil and water samples contained certain parasitic fungal phyla that could impact crop health and growth.

## Microbial genus level community structure and functional composition

In terms of function, chemoheterotrophy, aerobic chemoheterotrophy, and phototrophy are the main energy sources for the prokaryotic community in the eco-soil of the rice-crab symbiosis system. In the paddy soil and surrounding ditch soil, inorganic nitrogen fixation predominates, whereas organic nitrogen consumption is dominant. In contrast, carbon fixation in the paddy water and surrounding ditch water is primarily driven by photosynthesis, with a large

presence of chloroplasts serving as strong evidence. Rice cultivation is a significant source of methane ($CH_4$) emissions, a key agricultural greenhouse gas. In this study, we identified numerous bacteria closely related to methanogens, including *norank-f-Bacteroidetes-vadinHA17*, which had a relative abundance of 4.06% in DS samples, significantly higher than in other treatments ($p < 0.05$). This bacterium is known for degrading sawdust and polyacrylamide [24]. The relative abundance of *norank_o_SBR1031* was 3.98% in RS and 2.11% in DS groups, although little research has been conducted on this bacterium. It is an abundant genus during the cooling and maturation phases of composting, potentially aiding in nitrogen compound conversion [25]. *Norank_f_steroidobacteraceae* was positively correlated with soil $NO_3$-N concentrations [26] and contributes to carbon and nitrogen cycling through organic matter degradation, organic acid consumption, denitrification, and methane production [27]. Several genera with relative abundances between 1% and 2% were also found in the soil samples. The genus *norank_c_KD4-96* within the Chloroflexi phylum could promote the relative abundance of the Fe(II)-oxidizing group (*Gallionellaceae*, *Nitrospira*), which plays a role in iron absorption and nitrite oxidation [28]. Additionally, *Gemmatimonas* and *Sporomusaceae* are crucial for the degradation of phenanthrene and $CH_4$ production in paddy soils [29].

A large number of bacteria and fungi involved in carbohydrate degradation were found in both soil samples. Among bacteria, *norank_c_Dojkabacteria*, *norank_f_Prolixibacteraceae*, and *Smithella* showed similar trends. These microorganisms are members of acid- and acetate-producing populations, fermenting polysaccharides (such as cellobiose and starch) into acids, which can influence environmental pH [30]. *Marmoricola*, a Gram-positive bacterium, is widely found in marble, beach sediments, and forest soils and can assimilate various sugars and organic compounds such as D-glucose, phenylacetic acid, adipic acid, and biphenyl [31, 32]. Strains of *Marmoricola* isolated from wheat have been shown to degrade the fungal toxin deoxynivalenol (DON), which affects the growth of crops like rice, corn, and wheat. *Nocardioides* (2.39%) is another efficient DON-degrading genus, utilizing DON as its sole carbon and energy source under aerobic conditions and producing a new intermediate, 3-epi-DON, during the degradation process [33]. Notably, *Nocardioides*, known for its herbicide-degrading abilities [34], was distributed across multiple samples: RS (2.386%), DS (1.168%), PW (0.0466%), and SW (0.009532%). This genus can use a variety of organic substances as carbon sources, including petroleum hydrocarbons, aromatic compounds, and nitrogenous heterocycles. In addition to organic pollutant degradation, some *Nocardioides* strains are known to effectively degrade and transform steroids [35]. Furthermore, carbohydrate-fermenting bacteria, such as *norank_f_Anaerolineaceae*, degrade carbon-containing organic matter into $CO_2$ [36]. On the fungal side, an unclassified strain from the *Rozellomycota* phylum was observed. Large amounts of cow dung and straw biomass were decomposed into simple sugars and amino acids, providing nutrition for rice, with the coprophilous fungus *Sporormiella* being a prime example. *Pseudeurotium* is a common peatland fungus, and many species within this genus exhibit strong cellulolytic activity. Research has shown that biochar derived from crop residues (such as rice, wheat, and corn straw) can increase the relative abundance of beneficial fungi (*Pseudeurotium*), though excessive biochar levels may inhibit their growth [37]. Additionally, the genus *Pseudeurotium* plays a role in promoting crop growth and suppressing pathogens, and it can also degrade volatile organic compounds [38]. *Pseudeurotium* is more commonly found in traditional paddy soils without heavy metal contamination, while its abundance decreases in soils with significant heavy metal pollution [39].

In the water samples, *norank_f_norank_o_Chloroplast* and *unclassified_f_Rhodobacteraceae* were the dominant genera in the PW and SW samples. Research on *norank_o_Chloroplast* is limited, but Qiu's research identified it as a dominant genus in aquaculture sediments, showing a positive correlation with TAN and no correlation with TP or TN [40]. Yu et al. suggested

that it is a nitrogen-fixing bacterium [41]. *Rhodobacteraceae* are aquatic bacteria that include aerobic photoheterotrophs and purple non-sulfur bacteria that conduct photosynthesis in anaerobic environments [42]. They utilize a gene cluster for photosynthesis (PGC), which encodes all proteins necessary for anoxygenic photosynthesis, as well as enzymes, regulatory proteins, and cofactors involved in the biosynthesis pathways of bacteriochlorophyll and carotenoids [43].

The genus *LD26* was highly abundant in the water samples, particularly in the SW group, where it was the dominant genus. It belongs to the *Spartobacteria* class, and metagenomic analysis revealed its ability to utilize common carbon sources found in phytoplankton (cellulose, mannan, xylan, chitin, starch) and sulfated polysaccharides [44]. *Cyanobium PCC-6307* is a freshwater cyanobacterium commonly found in warm waters, serving as an indicator of potential algal bloom pollution [45]. Phototrophic bacteria, including cyanobacteria, play a key role in the energy cycling of both the paddy field and surrounding ditch water. However, the high abundance of cyanobacteria in the water also raises concerns about the risk of algal blooms.

In addition to functional microorganisms, pathogens are also a crucial part of the microbial community in the rice-crab symbiosis system. *Cladosporium delicatulum* from the *Cladosporium* genus and *Acremonium stromaticum* from the *Acremonium* genus are pathogens that infect rice, while antagonistic *Acremonium* species exhibit antibacterial, cytotoxic, phytotoxic, antiviral, insecticidal, and enzyme-inhibiting activities [46]. These two types of microorganisms help maintain the ecological balance in paddy fields. *Metschnikowia bicuspidata*, belonging to the *Spermophtoraceae* family, was primarily concentrated in the bottom soil of the experimental field ditches (28.81%), but its abundance in paddy soil was significantly lower (0.22%). This pathogen can cause milky disease in the Chinese mitten crab, leading to weight loss, reduced vitality, and the characteristic opaque, milky appearance of the muscle tissue and joint membranes of the legs and thoracic area [47]. Eventually, this condition can lead to organ failure and death [48]. It can also infect other aquatic organisms, such as the giant freshwater prawn and various fish species.

## Conclusion

In conclusion, our study demonstrates significant differences in bacterial and fungal diversity between soil and water samples, while no significant differences in microbial diversity were observed between the two soil samples and the two water samples. Dominant bacterial phyla (Proteobacteria, Cyanobacteria, Actinobacteria, Bacteroidetes, and Chloroflexi) and dominant fungal phyla (Ascomycota, Rozellomycota, Chytridiomycota, and Basidiomycota) primarily contribute to organic matter decomposition, secondary metabolite secretion, and autotrophic processes, thereby influencing their respective habitats. Additionally, we identified and focused on microorganisms such as *Metschnikowia* and *Nocardioides*, which play a significant role in shaping the biochemical environment of the rice-crab symbiotic system, providing essential support for the healthy development of integrated agriculture and aquaculture.

## Supporting information

**S1 File.**
(ZIP)

## Author Contributions

**Conceptualization:** Tian Juncang, Wang Zhi.

**Data curation:** Guo Yang.

**Funding acquisition:** Tian Juncang.

**Investigation:** Guo Yang.

**Methodology:** Tian Juncang, Wang Zhi.

**Project administration:** Tian Juncang.

**Supervision:** Tian Juncang.

**Writing – original draft:** Guo Yang.

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
