## [Decision Letter · Decision Letter 0]

23 Sep 2024

PONE-D-24-35167Comparative Analysis of Soil and Water Microbial Community Structures and Functions in the Rice-Crab Symbiotic System Based on High-throughput SequencingPLOS ONE

Dear Dr. Yang,

Thank you for submitting your manuscript to PLOS ONE. After careful consideration, we feel that it has merit but does not fully meet PLOS ONE’s publication criteria as it currently stands. Therefore, we invite you to submit a revised version of the manuscript that addresses the points raised during the review process.

We look forward to receiving your revised manuscript.

Kind regards,

Sayed Haidar Abbas Raza

Academic Editor

PLOS ONE

“National Key Research and Development Project (No. 2021YFD1900605-04)

First-class discipline of Ningxia High Education Institutions (Water Engineering Discipline) funded project (Grant No. NXYLXK2021A03).”

“No.”

5. We note that you have indicated that there are restrictions to data sharing for this study. PLOS only allows data to be available upon request if there are legal or ethical restrictions on sharing data publicly. For more information on unacceptable data access restrictions, please see http://journals.plos.org/plosone/s/data-availability#loc-unacceptable-data-access-restrictions.

6. PLOS requires an ORCID iD for the corresponding author in Editorial Manager on papers submitted after December 6th, 2016. Please ensure that you have an ORCID iD and that it is validated in Editorial Manager. To do this, go to ‘Update my Information’ (in the upper left-hand corner of the main menu), and click on the Fetch/Validate link next to the ORCID field. This will take you to the ORCID site and allow you to create a new iD or authenticate a pre-existing iD in Editorial Manager.

Reviewers' comments:

Reviewer's Responses to Questions

**Comments to the Author**

1. Is the manuscript technically sound, and do the data support the conclusions?

Reviewer #1: Partly

Reviewer #2: Yes

2. Has the statistical analysis been performed appropriately and rigorously? 

Reviewer #1: Yes

Reviewer #2: Yes

3. Have the authors made all data underlying the findings in their manuscript fully available?

Reviewer #1: Yes

Reviewer #2: No

4. Is the manuscript presented in an intelligible fashion and written in standard English?

Reviewer #1: Yes

Reviewer #2: Yes

5. Review Comments to the Author

Reviewer #1: 1. There is need of a revision regarding English language. Mycological terms are also suggested when discussing about the fungal communities in results.

2. There is lacking of appropriate reference citations in material method portion.

3. DNA isolation and Sequencing portion need a little more elaboration to clear methodology.

4. I am unable to find any result about "Functions in the Rice-Crab Symbiotic System Based on High-throughput Sequencing" that is the main part of research title.

5. A thorough editing of the article is suggested

6. References should also be in well formatted fashion.

Reviewer #2: The manuscript "Comparative Analysis of Soil and Water Microbial Community Structures and Functions in the Rice-Crab Symbiotic System Based on High-throughput Sequencing" studies the microbial diversity in the Rice-Crab ecosystem. The activities of crabs not only help to loosen the soil and control weeds in rice paddies, but their excrement can be used as natural fertilizer for rice, effectively reducing the use of chemical fertilizers and pesticides. Rice gives crabs the cool environment and plenty of food resources they need for growth. By breaking down organic matter in the soil, taking part in the mineralization process, influencing the use of soil nutrients, and impacting the biogeochemical cycling of elements, soil microorganisms affect the growth, quality, and yield of rice. The microbial community also plays a significant role in controlling the water's environment and achieving water purification, which when combined with rice and crabs creates a balanced ecology. In this study the authors performed microbial sequencing based on the Illumina sequencing platform on rice inter-root soil, soil at the bottom of the ring ditch, water samples from the ring ditch, and water samples from the paddy field in the rice-crab aquaculture area, and analyzed the abundance, composition, and diversity of bacterial and fungal diversity at different sites in the rice-crab symbiotic environment. They identified a total of 54 bacterial phyla and 1227 genera across the four samples. Fungal diversity was relatively low in all samples compared to bacteria, with only nine phyla and 97 genera detected in all samples. Based on biomarkers the authors found significant differences in microbial composition between sampling sites. Microbiological analysis of rice soil, ditch bottom soil, paddy water body, and ring ditch water body revealed that higher chlorophyll and photosynthetic characteristics during the rice reproductive period showed significant differences in bacterial and fungal diversity between soil and water samples, whereas there was no significant difference in the diversity of the two soils versus the two water bodies. This manuscript is well written. The authors have explained the methods in detail and the conclusions are based on the results but I have a few concerns

1. The authors can prepare a figure showing the Rice-Crab ecosystem and the regions from which the samples were collected. This figure will be very valuable to the readers as then the readers will be able to directly correlate the findings with the regions in the ecosystem.

2. All the data is not freely available. The authors need to be very clear about which data is available and which is not.

3. Some of the abbreviations are used without providing the full form ex OTU.

4. OTU is misspelled as OUT at many places including Pg 16 and Pg 17

6. PLOS authors have the option to publish the peer review history of their article (what does this mean?). If published, this will include your full peer review and any attached files.

Reviewer #1: **Yes: **Dr. Muhammad Ali

Reviewer #2: No

---

## [Author Response · Author response to Decision Letter 0]

30 Oct 2024

Response to Essay Revision Comments

Dr. Muhammad Ali, 

I hope this message finds you well. I would like to sincerely thank you for taking the time to review my essay and providing such detailed and insightful feedback. I truly appreciate your suggestions, which I believe will greatly improve the quality of my work.

In response to your comments, I have addressed the following key points:

1. Language Clarity: I have thoroughly revised the entire text to improve the flow and readability, ensuring a smoother reading experience.

2. Materials and Methods: I have incorporated the steps for specific sequencing analysis, as you suggested, and believe these changes effectively address the support issues. 

3. References: Based on your recommendation, I have updated the citation format, added references to the *Materials and Methods* section, and included more supporting evidence throughout the essay. 

Additionally, I reviewed the entire essay to ensure that the revisions align with your overall feedback. If there are any other adjustments you'd like me to make or areas requiring further clarification, I would be happy to discuss them or make additional revisions.

Once again, I am deeply grateful for your valuable input, and I hope the revised version of the essay meets your expectations. I look forward to your feedback.

Warm regards, 

Yang Guo 

Mr. 

nxuguoyang2024@163.com

---

## [Decision Letter · Decision Letter 1]

13 Nov 2024

PONE-D-24-35167R1Composition and functional diversity of soil and water microbial communities in the rice-crab symbiosis systemPLOS ONE

Dear Dr. Yang,

Thank you for submitting your manuscript to PLOS ONE. After careful consideration, we feel that it has merit but does not fully meet PLOS ONE’s publication criteria as it currently stands. Therefore, we invite you to submit a revised version of the manuscript that addresses the points raised during the review process.

We look forward to receiving your revised manuscript.

Kind regards,

Sayed Haidar Abbas Raza

Academic Editor

PLOS ONE

Journal Requirements:

Reviewers' comments:

Reviewer's Responses to Questions

**Comments to the Author**

1. If the authors have adequately addressed your comments raised in a previous round of review and you feel that this manuscript is now acceptable for publication, you may indicate that here to bypass the “Comments to the Author” section, enter your conflict of interest statement in the “Confidential to Editor” section, and submit your "Accept" recommendation.

Reviewer #2: (No Response)

2. Is the manuscript technically sound, and do the data support the conclusions?

Reviewer #2: Yes

3. Has the statistical analysis been performed appropriately and rigorously? 

Reviewer #2: Yes

4. Have the authors made all data underlying the findings in their manuscript fully available?

Reviewer #2: Yes

5. Is the manuscript presented in an intelligible fashion and written in standard English?

Reviewer #2: Yes

6. Review Comments to the Author

Reviewer #2: The authors have not responded to my concerns directly. Though I can see that they have incorporated the suggestions in the manuscript, they have not addressed the concerns directly. I suggest the authors to directly comment point by point to each concern.

7. PLOS authors have the option to publish the peer review history of their article (what does this mean?). If published, this will include your full peer review and any attached files.

Reviewer #2: No

---

## [Author Response · Author response to Decision Letter 1]

8 Dec 2024

Modified according to the editing requirements

---

## [Decision Letter · Decision Letter 2]

15 Dec 2024

PONE-D-24-35167R2Composition and functional diversity of soil and water microbial communities in the rice-crab symbiosis systemPLOS ONE

Dear Dr. Yang,

Thank you for submitting your manuscript to PLOS ONE. After careful consideration, we feel that it has merit but does not fully meet PLOS ONE’s publication criteria as it currently stands. Therefore, we invite you to submit a revised version of the manuscript that addresses the points raised during the review process.

We look forward to receiving your revised manuscript.

Kind regards,

Sayed Haidar Abbas Raza

Academic Editor

PLOS ONE

Journal Requirements:

Reviewers' comments:

Reviewer's Responses to Questions

**Comments to the Author**

1. If the authors have adequately addressed your comments raised in a previous round of review and you feel that this manuscript is now acceptable for publication, you may indicate that here to bypass the “Comments to the Author” section, enter your conflict of interest statement in the “Confidential to Editor” section, and submit your "Accept" recommendation.

Reviewer #2: (No Response)

2. Is the manuscript technically sound, and do the data support the conclusions?

Reviewer #2: Yes

3. Has the statistical analysis been performed appropriately and rigorously? 

Reviewer #2: Yes

4. Have the authors made all data underlying the findings in their manuscript fully available?

Reviewer #2: Yes

5. Is the manuscript presented in an intelligible fashion and written in standard English?

Reviewer #2: Yes

6. Review Comments to the Author

Reviewer #2: The authors have not replied to my concerns. Though we have asked the authors last time also to provide point by point answers to the concerns. The authors should understand that they have to provide written answers to each concern before the manuscript can be accepted. Even if they have performed the changes in the manuscript, they still have to provide the answers to the concerns.

Please provide answers to the following concerns. For example for the first concern the authors can write that they have prepared this figure.

Concerns

1. The authors can prepare a figure showing the Rice-Crab ecosystem and the regions from which the samples were collected. This figure will be very valuable to the readers as then the readers will be able to directly correlate the findings with the regions in the ecosystem.

2. All the data is not freely available. The authors need to be very clear about which data is available and which is not.

3. Some of the abbreviations are used without providing the full form ex OTU.

4. OTU is misspelled as OUT at many places including Pg 16 and Pg 17

7. PLOS authors have the option to publish the peer review history of their article (what does this mean?). If published, this will include your full peer review and any attached files.

Reviewer #2: No

---

## [Author Response · Author response to Decision Letter 2]

15 Dec 2024

Format and Style:

I have adjusted the article's layout according to the formatting guidelines you provided, including heading levels, paragraph spacing, font size, etc., to meet publication standards. I have checked and supplemented the complete form of the OTU in the article.

Images and Citations:

I have made a chart showing the rice crab ecosystem and the areas where the samples were collected. As per your request, I have replaced some images to ensure all are copyright-clear. I have uploaded the graphic files to the Preflight Analysis and Conversion Engine (PACE) and ensured that the graphics meet the requirements.

Data：

All data I have added to the attachment is free.

References：

For all cited literature, I have verified and updated the latest publication information.

---

## [Decision Letter · Decision Letter 3]

18 Dec 2024

Composition and functional diversity of soil and water microbial communities in the rice-crab symbiosis system

PONE-D-24-35167R3

Dear Dr. Yang,

We’re pleased to inform you that your manuscript has been judged scientifically suitable for publication and will be formally accepted for publication once it meets all outstanding technical requirements.

Kind regards,

Sayed Haidar Abbas Raza

Academic Editor

PLOS ONE

Additional Editor Comments (optional):

Reviewers' comments:

Reviewer's Responses to Questions

**Comments to the Author**

1. If the authors have adequately addressed your comments raised in a previous round of review and you feel that this manuscript is now acceptable for publication, you may indicate that here to bypass the “Comments to the Author” section, enter your conflict of interest statement in the “Confidential to Editor” section, and submit your "Accept" recommendation.

Reviewer #2: All comments have been addressed

2. Is the manuscript technically sound, and do the data support the conclusions?

Reviewer #2: Yes

3. Has the statistical analysis been performed appropriately and rigorously? 

Reviewer #2: Yes

4. Have the authors made all data underlying the findings in their manuscript fully available?

Reviewer #2: Yes

5. Is the manuscript presented in an intelligible fashion and written in standard English?

Reviewer #2: Yes

6. Review Comments to the Author

Reviewer #2: The authors have responded to all the concerns. They have provided point by point answers to all the concerns.

7. PLOS authors have the option to publish the peer review history of their article (what does this mean?). If published, this will include your full peer review and any attached files.

Reviewer #2: No

---

## [Editor Report · Acceptance letter]

30 Dec 2024

PONE-D-24-35167R3 

PLOS ONE

Dear Dr. Yang, 

I'm pleased to inform you that your manuscript has been deemed suitable for publication in PLOS ONE. Congratulations! Your manuscript is now being handed over to our production team.

Kind regards, 

on behalf of

Dr. Sayed Haidar Abbas Raza 

Academic Editor

PLOS ONE